# Symmetric-Key-Based Authentication among the Nodes in a Wireless Sensor and Actuator Network

**DOI:** 10.3390/s22041403

**Published:** 2022-02-11

**Authors:** Thibaut Vandervelden, Ruben De Smet, Kris Steenhaut, An Braeken

**Affiliations:** 1Department of Engineering Sciences and Technology (INDI), Vrije Universiteit Brussel (VUB), 1050 Brussels, Belgium; an.braeken@vub.be; 2Department of Electronics and Informatics (ETRO), Vrije Universiteit Brussel (VUB), 1050 Brussels, Belgium; rdesmeta@etrovub.be (R.D.S.); ksteenha@etrovub.be (K.S.)

**Keywords:** wireless sensor networks, authentication, nonrepudiation, cooperative network, anonymity, TESLA

## Abstract

To enable today’s industrial automation, a significant number of sensors and actuators are required. In order to obtain trust and isolate faults in the data collected by this network, protection against authenticity fraud and nonrepudiation is essential. In this paper, we propose a very efficient symmetric-key-based security mechanism to establish authentication and nonrepudiation among all the nodes including the gateway in a distributed cooperative network, without communicating additional security parameters to establish different types of session keys. The solution also offers confidentiality and anonymity in case there are no malicious nodes. If at most one of the nodes is compromised, authentication and nonrepudiation still remain valid. Even if more nodes get compromised, the impact is limited. Therefore, the proposed method drastically differs from the classical group key management schemes, where one compromised node completely breaks the system. The proposed method is mainly based on a hash chain with multiple outputs defined at the gateway and shared with the other nodes in the network.

## 1. Introduction

Wireless Sensor and Actuator Networks (WSANs) are currently very popular and are nowadays applied in a multitude of domains for monitoring and control. Application domains are, for instance, industrial automation, home automation, agriculture, transportation and logistics, body area networks, military applications, underground sensor networks, traffic monitoring, and air pollution monitoring. WSANs consist of a group of sensors, measuring different types of environmental parameters such as temperature, sound, humidity, and motion together with actuators equipped with, e.g., servos and motors that interact with them.

A very important requirement in WSANs is to present trustworthy measurements to the end users such that they can have confidence and accept conclusions drawn from them. At the same time, a long lifetime should be guaranteed with minimum to no maintenance overhead as the sensors are mostly battery powered and maintenance is very expensive. Therefore, adequate and efficient security mechanisms should be included, which are in the first place focused on the establishment of authentication and integrity.

Since efficiency is of utmost importance, symmetric-key-based solutions should be addressed. A very popular and elegant symmetric-key-based broadcast authentication mechanism for WSANs, called TESLA [1], was developed in 2002. However, this protocol only allows authentication by the root, also known as sink or cluster head, to the other nodes of the network. An effective symmetric-key-based mechanism, which also addresses authenticity, coming from the different nodes of a distributed network (without predefined positions) has not yet been proposed in literature as far as the authors are aware. This shortcoming has been circumvented till now by the use of a common group key shared among all nodes in the network. It is not the most optimal and secure solution as it does not provide protection against malicious insiders, who can take the identity of the others without them being notified, and thus only offers authentication at the group level. Moreover, as soon as one of the nodes is captured and corrupted, the whole security of the network is broken.

## 2. Related Work

There exists a wide variety of solutions addressing key management in wireless sensor networks for different types of keys (pairwise keys, group keys, individual keys) for different types of network topologies (hierarchical, flat). Both symmetric- and asymmetric-key-based approaches have been proposed. For efficiency reasons, we focus on the most efficient ones—those being the symmetric-key-based ones. We also limit our discussion to key management schemes for WSANs, which address security at the group level, and exclude schemes such as SPINS [2] and BROSK [3] whose focus is on the construction of pairwise keys.

One of the most complete symmetric key management schemes for WSANs is the Localized Encryption and Authentication Protocol (LEAP) [4]. This protocol for hierarchical networks explains how to establish a pairwise key between two cluster heads, a group key in the same cluster, and a network key.

Other symmetric-key-based group key management schemes for tree- and star-based networks can be found in [5,6,7]. In [5], the group key is constructed by means of the evaluation of a bivariate polynomial in combination with Lagrange interpolation. In [6,7], the group key is constructed using a generator matrix.

For distributed ad-hoc networks with randomly deployed nodes, there are only a limited number of symmetric-key-based proposals for key management since it is more difficult as the positions of the nodes in the network are not known in advance. In [8], the keys are generated by relying on a robot-assisted network bootstrapping technique. Recently, in [9], another approach has been proposed to derive multicast keys among a dynamic group of nodes by using a cloud-based Network Multicast Manager (NMM) and a dedicated asymmetric distribution of key material between NMM and a sensor node.

However, in all of these approaches, the focus is on the generation of a group key, sometimes combined with a pairwise key between two nodes or an individual key with the cluster node/sink. The authentication of the individual sensor nodes verifiable by each of the other members in the group has not yet been addressed via a cryptographic approach. If it is considered at all, trust- and reputation-based schemes are defined [10], which typically require very high computational efforts and are mainly installed at the gateway level.

Despite the needed efforts, the feature is worth considering, particularly for cooperative WSANs [11]. First, it allows the node to verify the trustworthiness of its neighbor nodes such that the data received from them can be used for own purposes (e.g., calibration). Second, it also enables to localize malicious nodes in the network and to avoid all types of corresponding attacks such as wormholes, sinkhole, Sybil, sleep, selective forward, and denial of service. For the detection of malicious nodes, several statistical [12] and AI-based methods [13] have been explained in literature. However, a solution based on a cryptographic approach is not yet available in literature, although it is one of the most effective assets to avoid these type of attacks.

For the authentication of the cluster node using symmetric key cryptography, the TESLA protocol [1] and its variants [14,15,16] offer currently the most efficient solution. These schemes are based on a hash chain and on an orchestrated release of each of the values of the chain in predefined timeslots. As a consequence, there is a delay of one timeslot to verify the authenticity of the message. Note that there is also the scheme of [17], which also uses a hash chain, but now combined with authenticated encryption resulting in a scheme where there is no delay for authentication. The proposed scheme was applied in an edge-based architecture.

In this paper, we extend the usage of the TESLA protocol to all the sensor nodes in the network by relying on the extendable output feature of the sponge functions, proposed by Bertoni et al. in [18].

## 3. Preliminaries

First, we provide some background information on the network model and the corresponding attacker model. Next, the required security features are elaborated and a short description of the cryptographic operations used in our system is given.

### 3.1. Network Model

We consider a distributed WSAN, consisting of constrained sensors with limited energy and computational resources. The sensors can change position in the network (e.g., animal monitoring) or can stay at fixed positions (e.g., pollution monitoring at different predefined locations). We assume that changes in the network are limited. Nodes can be added, but if nodes are removed, then the memory on it should also be erased.

The gateway sends a broadcast message to all the sensor nodes in the network, which can reply by sending a message back to the gateway. Both communications can be done via a multihop approach. Each of the nodes on the routing path should be able to derive the content and origin of the message and verify its authenticity and integrity. A representation of such a network is depicted in Figure 1.

### 3.2. Attacker Model

We make the following assumptions regarding the attack model.

The attacker is active during the whole process and is able to eavesdrop the messages sent over the wireless channel. The attacker can actively change, delete, alter, and replay messages (or parts of it) sent over the channel.The gateway is considered trusted and possesses the required protection measures to resist security attacks. As a minimum, secure storage of key material is guaranteed.The security material in the nodes is stored in tamper-proof memory, such that it is difficult for an attacker to reveal it.

### 3.3. Security Features

The classical security features of confidentiality, integrity, authentication, and anonymity together with nonrepudiation are attained at group level. This means that all nodes in the network are able to send messages satisfying these features as long as no network node is malicious. Compared with other schemes, we make it possible to address these features among all nodes, without storage of a shared pairwise key upfront or submission of random values to derive session keys during the actual operation.

The strong point in our scheme is that the security features of authentication, integrity, and nonrepudiation also apply on an individual level and are maintained as long as at most one of the nodes gets compromised. In schemes based on group keys, a malicious node can take the identity of another node of the group, which is not possible in our scheme. Our scheme protects against impersonation and man-in-the-middle attacks as long as at most one node is hijacked.

In case more than one node is compromised, the authentication and integrity of the gateway is still valid but they are impacted for a limited number of nodes, which vary in the different timeslots. Therefore, it also enables fast fault isolation and effective intrusion detection and avoidance.

### 3.4. Cryptographic Operations

Taking into account the limited computational resources and the minimum lifetime of three years to be guaranteed by the sensors, the overhead created due to ensuring the required security features should be as low as possible. Therefore, the cryptographic operations used in the proposed scheme are limited to xor operations, encryptions, and hash operations.

We denote the xor operation by ⊕, the symmetric key encryption by means of a common shared key *K* by $E_{K}\left(\right)$ and corresponding decryption by $D_{K}\left(\right)$. We assume the usage of an Authenticated Encryption (AE) mode such as AES-GCM or AES-CCM.

The hash operation used in our scheme is based on the SHA3 standard SHAKE256(*M*,*d*), which is an algorithm providing an output with variable length *d* and providing a minimum security strength of min(d/2, 256)-bit against collision attacks and min(*d*, 256)-bit against preimage and second preimage attacks. We denote by $H_{i}\left(M\right)=\left(M_{1},M_{2},\ldots ,M_{i}\right)$ the SHAKE256(*M*,*d*) algorithm working on message *M* providing *i* output blocks, where the total length of all blocks in the output vector equals to *d*, $|M_{1}\left|+\right|M_{2}\left|+\ldots +\right|M_{i}|=d$.

## 4. Proposed Scheme

The scheme consists of three different entities: nodes, gateway, and application server. We consider two different phases: key initialization and actual communication and authentication.

### 4.1. Key Initialization

Suppose there are $2^{n}-1$ nodes with identities $ID_{i}=i\in \left\{1,\;\ldots ,\;2^{n}-1\right\}$ in the WSAN. The gateway chooses two random keys $K^{0}$ and $K_{e}^{0}$ and computes a hash chain of both with length *t* and a corresponding output consisting of n+1 blocks and 1 block using the hash functions $H_{n+1}$ and $H_{1}$, respectively. The hash chain of key $K^{0}$ will be mainly used to derive material for authentication purposes and the hash chain of key $K_{e}^{0}$ will define the encryption keys in the process. There are *n* independent components needed in the hash chain of $K^{0}$ in order to be able to define unique authentication material for each of the $2^{n}-1$ nodes in the network.
$$\begin{array}{ccc} j=1 & : & K^{1}=H_{n+1}\left(K^{0}\right)=\left(K_{0}^{1},K_{1}^{1},K_{2}^{1},\ldots ,K_{n}^{1}\right),K_{e}^{1}=H_{1}\left(K_{e}^{0}\right)\\ j=2 & : & K^{2}=H_{n+1}\left(K_{0}^{1}\right)=\left(K_{0}^{2},K_{1}^{2},K_{2}^{2},\ldots ,K_{n}^{2}\right),K_{e}^{2}=H_{1}\left(K_{e}^{1}\right)\\ j=3 & : & K^{3}=H_{n+1}\left(K_{0}^{2}\right)=\left(K_{0}^{3},K_{1}^{3},K_{2}^{3},\ldots ,K_{n}^{3}\right),K_{e}^{3}=H_{1}\left(K_{e}^{2}\right)\\ & \vdots & \\ j=t & : & K^{t}=H_{n+1}\left(K_{0}^{t-1}\right)=\left(K_{0}^{t},K_{1}^{t},K_{2}^{t},\ldots ,K_{n}^{t}\right),K_{e}^{t}=H_{1}\left(K_{e}^{t-1}\right) \end{array}$$
Note that for all j∈{1,…,t}, the variables $K_{0}^{j},K_{e}^{j}$ consist of 256 bits and $K_{s}^{j}$ of 128 bits for all s∈{1,…,n}. We also denote the first 128 bits of the variables $K_{0}^{j}$ and $K_{e}^{j}$ by ${\bar{K}}_{0}^{j}$ and ${\bar{K}}_{e}^{j}$.

Each node *i* receives $t+1,K_{0}^{t+1}=H_{1}\left(K_{0}^{t}\right)$ installed, corresponding with the root of the first hash chain.

Define for each node *i* and timeslot j∈{1,…,t} the parameter *b*, which equals to
$$\begin{array}{ccc} b & = & \left(i+j\;\left(\mathrm{mod}\;2^{n}-1\right)\right)+1\;\;\;\;\;\;if\;\;\;i+j\neq 2^{n}-1\\ & = & 1\;\;\;\;\;\;\;\;\;\;\;\;\;\;\;\;\;\;\;\;\;\;\;\;\;\;\;\;\;\;\;\;\;\;\;\;\;\;\;\;\;\;\;\;\;if\;\;\;i+j=2^{n}-1 \end{array}$$

Denote the binary representation of *b* by $b=(b_{1},\ldots ,b_{n})$. Then, the node *i* also obtains for each timeslot j∈{1,…,t} the following tuple $K_{ij}=\left(K_{i}^{j1},K_{i}^{j2}\right)$ installed:$$\begin{array}{ccc} K_{i}^{j1} & = & {⊕}_{s=1}^{n}b_{s}K_{s}^{j}\\ K_{i}^{j2} & = & E_{{\bar{K}}_{0}^{j+1}⊕K_{i}^{j1}}\left({\bar{K}}_{e}^{j}\right) \end{array}$$

Note that the first variable in the tuple $K_{i}^{j1}$ corresponds with a linear combination, defined by *b*, of the hash outputs of $K_{0}^{j-1}$. Since *b* varies for each node *i* and timestamp *j*, the overall complexity of the linear combinations can be considered similar for all nodes. The second variable $K_{i}^{j2}$ enables deriving the first part of the hash chain of $K_{e}^{0}$.

The key $K_{i}^{j1}$ is used in the later protocol to firstly verify the authentication of the hash chain of the gateway and to secondly authenticate its own message, which can be publicly verified by the other nodes in the next timestamp. The key $K_{i}^{j2}$ enables the derivation of the decryption key of the encrypted message coming from the gateway, which is also used as the group encryption key by the nodes in the same timestamp. Note that this key ${\bar{K}}_{e}^{j}$ can only be defined after reception of the value $K_{0}^{j+1}$, as this is required to decrypt $K_{i}^{j2}$.

The gateway installs the hash chain $K^{0},K_{0}^{1},\;\ldots ,\;K_{0}^{t+1}$ and $K_{e}^{0},K_{e}^{1},\;\ldots ,\;K_{e}^{t}$. We consider nodes and gateway synchronized, which is also needed for several well-known medium access and radio duty cycling (RDC) protocols such as IEEE 802.15.4e Time Slotted Channel Hopping (TSCH).

**Example** **1.**
*The construction of the key material for n=2,t=4 and node $ID_{1}$ (i=1) is as follows.*

*Construction of key material, starting from two root keys $K^{0}$ and $K_{e}^{0}$.*

$$\begin{array}{ccc} j=1 & : & K^{1}=H_{3}\left(K^{0}\right)=\left(K_{0}^{1},K_{1}^{1},K_{2}^{1}\right),K_{e}^{1}=H_{1}\left(K_{e}^{0}\right)\\ j=2 & : & K^{2}=H_{3}\left(K_{0}^{1}\right)=\left(K_{0}^{2},K_{1}^{2},K_{2}^{2}\right),K_{e}^{2}=H_{1}\left(K_{e}^{1}\right)\\ j=3 & : & K^{3}=H_{3}\left(K_{0}^{2}\right)=\left(K_{0}^{3},K_{1}^{3},K_{2}^{3}\right),K_{e}^{3}=H_{1}\left(K_{e}^{2}\right)\\ j=4 & : & K^{4}=H_{3}\left(K_{0}^{3}\right)=\left(K_{0}^{4},K_{1}^{4},K_{2}^{4}\right),K_{e}^{4}=H_{1}\left(K_{e}^{3}\right) \end{array}$$


*For $ID_{1}$, the following material is stored.*

$$\begin{array}{ccc} j=1(b=3) & : & K_{1}^{11}=K_{1}^{1}⊕K_{2}^{1},K_{1}^{12}=E_{{\bar{K}}_{0}^{2}⊕K_{1}^{11}}\left({\bar{K}}_{e}^{1}\right)\\ j=2(b=1) & : & K_{1}^{21}=K_{1}^{2},K_{1}^{22}=E_{{\bar{K}}_{0}^{3}⊕K_{1}^{21}}\left({\bar{K}}_{e}^{2}\right)\\ j=3(b=2) & : & K_{1}^{31}=K_{2}^{3},K_{1}^{32}=E_{{\bar{K}}_{0}^{3}⊕K_{1}^{31}}\left({\bar{K}}_{e}^{3}\right)\\ j=4(b=3) & : & K_{1}^{41}=K_{1}^{4}⊕K_{2}^{4},K_{1}^{42}=E_{{\bar{K}}_{0}^{4}⊕K_{1}^{41}}\left({\bar{K}}_{e}^{4}\right)\\ j=5\;\;\;\;\;\;\;\;\; & : & K_{0}^{5}=H_{1}\left(K_{0}^{4}\right) \end{array}$$



### 4.2. Actual Communication and Authentication Process

In each timeslot *j*, the gateway starts with a broadcast message and the nodes react to this message. Both steps are explained in more detail below.

#### 4.2.1. Broadcast Message of Gateway

Denote by $M^{j}$ the data to be broadcasted by the gateway at timeslot j∈{t−1,…,1} with decreasing timeslots. The message $M^{j}$ can include different types of information. For instance, it can contain the list of nodes from whom data are (not) received, the list of malicious nodes, and data on the clock to enable synchronization. For each timeslot j∈{t−1,…,1}, the following information is broadcasted:$$\begin{array}{ccc} & & j,K_{0}^{j+1},E_{{\bar{K}}_{e}^{j}}\left(M^{j},H_{1}\left(M^{j},K_{0}^{j}\right)\right) \end{array}$$

#### 4.2.2. Reaction of Nodes

Upon arrival of this message, the sensors in the network perform the following actions.

**Step****1:** First, the nodes are able to verify the authenticity of the gateway, but not the real authenticity of the received broadcast message of the gateway.In order to do this, the authenticity of the received value $K_{0}^{j+1}$ is verified by computing the hash of it. In case j=t−1, this corresponds with the prestored value. If j<t−1, comparison is made with the previously sent value and the prestored key material of slot *j*.**Step****2:** Next, by construction of the key material stored in each node and the received value $K_{0}^{j+1}$, the node is able to extract ${\bar{K}}_{e}^{j}=D_{{\bar{K}}_{0}^{j+1}⊕K_{i}^{j1}}\left(K_{i}^{j2}\right)$ and decrypt the last part of the broadcast message.This key is also used to send, in the timeslot *j*, their own data $M_{i}^{j}$ in the same way, by encoding it in the following format:
$$\begin{array}{ccc} & & j,E_{{\bar{K}}_{e}^{j}}\left(i,M_{i}^{j},H_{1}\left(M_{i}^{j},K_{i}^{(j+1)1}\right)\right) \end{array}$$Note that using the same key material, multiple messages can be sent by the nodes during this timeslot. An enumeration of the order can be included in the message. The number of messages that can be received in each slot depends on the receivers’ buffer because messages are tentatively accepted based on the validation of the first part of the message and completely accepted on the next slot if the authentication tag is correct.**Step****3:** In the same timeslot, the node is able to extract the received message $M_{r}^{j}$ of all of its children or its neighbors (depending on the routing protocol) with identity *r* using the same decryption key ${\bar{K}}_{e}^{j}$. Again, this is an authenticity check at the group level. To enable it at the node level, the messages need to be stored such that they can be verified in the next timeslot.The sensors also store the broadcast message $\left(j,M^{j},H_{1}\left(M^{j},K_{0}^{j}\right),K_{0}^{j+1}\right)$ for further verification in the next timeslot.**Step****4:** Finally, the nodes can start verifying the individual authenticity of the messages stored from the previous timeslot j+1 coming from the gateway and the children or neighboring nodes using the received value $K_{0}^{j+1}$.

In timeslot *j*, the gateway is also collecting all the messages from the nodes and is able to check the validity of the messages at the node level as the gateway possesses the required security material.

**Example** **2.**
*The actual communication and authentication process for node $ID_{1}$ in the case n=2 at timeslot j=2 consists of the following steps, based on the key material as described in Example 1.*

$$\begin{array}{ccc} Step0 & : & Gatewaysends:2,K_{0}^{3},E_{{\bar{K}}_{e}^{2}}\left(M^{2},H_{1}\left(M^{2},K_{0}^{2}\right)\right)\\ Step1 & : & Node:H_{3}\left(K_{0}^{3}\right)=\left(K_{0}^{4},K_{1}^{4},K_{2}^{4}\right),andverifiesK_{0}^{4},K_{1}^{41}\\ Step2 & : & Node:{\bar{K}}_{e}^{2}=D_{{\bar{K}}_{0}^{3}⊕K_{1}^{21}}\left(K_{1}^{22}\right),\\ & & D_{{\bar{K}}_{e}^{2}}\left(M^{2},H_{1}\left(M^{2},K_{0}^{2}\right)\right),E_{{\bar{K}}_{e}^{2}}\left(2,M_{1}^{2},H_{1}\left(M_{1}^{2},K_{1}^{31}\right)\right)\\ Step3 & : & Nodedecrypts:{\left(E_{{\bar{K}}_{e}^{2}}\left(2,M_{r}^{2},H_{1}\left(M_{r}^{2},K_{r}^{31}\right)\right)\right)}_{r}\\ & : & Nodestores:\{2;\left(M^{2},H_{1}\left(M^{2},K_{0}^{2}\right)\right),{\left(M_{r}^{2},H_{1}\left(M_{r}^{2},K_{r}^{31}\right)\right)}_{r}\}\\ Step4 & : & Nodechecks:\left\{3;M^{3},H_{1}\left(M^{3},K_{0}^{3}\right)\right),{\left(M_{r}^{3},H_{1}\left(M_{r}^{3},K_{r}^{41}\right)\right)}_{r}\} \end{array}$$



#### 4.2.3. Addition/Removal of Nodes

Corresponding with the hash function $H_{n+1}$, there is security material for $2^{n}-1$ nodes. As long as there are less than $2^{n}-1$ nodes in the field, additional nodes can still securely join the network by storing the security material in the node.

In case a node needs to be removed, the memory on it should be completely erased as a malicious insider would be able to break the confidentiality and the anonymity of the consecutive communication (cf. Theorem 2).

We have assumed in the attacker model that the keys are stored in tamper-proof memory; therefore, an attacker will not be able to simply extract the key material of the node. The problem occurs in the case of so-called insider attacks, where nodes become malicious. In this situation, the node’s identity should be noted. If more potential malicious nodes pop up, a list of untrusted inputs should be made, based on the identities of the malicious nodes (cf. Theorem 3).

## 5. Security Evaluation

For the security proofs, we consider the usage of building blocks that are semantically secure under chosen plaintext attacks. As a consequence, the generated ciphertext and authentication tag can be considered indifferentiable from the ones generated by a random permutation or random function, respectively. To be more specific, in this proposed scheme, the selected algorithms SHAKE256(*M*,*d*) and AES satisfy this feature.

**Theorem** **1.**
*The protocol is able to offer confidentiality, integrity, authentication, anonymity, and nonrepudiation with a minimum security level of 128 bits with the condition that there are no malicious nodes in the network and the underlying security algorithms are not broken.*


**Proof.** First, we prove the statement from the gateway point of view, where the focus is on authentication, integrity, and confidentiality. A valid key $K_{0}^{j+1}$ in the broadcast message cannot be constructed by an attacker if the underlying hash algorithm offers protection against collision and preimage attacks, which has, in the best situation, a probability of $2^{-128}$ in the case of the SHAKE256(*M*,*d*) protocol. Without knowledge of the value ${\bar{K}}_{e}^{j}$, which is impossible to derive without having additional information (such as the trusted nodes) the broadcast message cannot be decrypted and is thus protected against confidentiality and integrity with a probability of $2^{-128}$, corresponding with a brute force guess of the key. In addition, due to the security of the building blocks under chosen plaintext attacks, the generated ciphertexts cannot be differentiated from a random value.For the trusted nodes, the transmitted message cannot be decrypted by attackers eavesdropping on the channel as they are encrypted with the same key used by the gateway; thus, they have a probability of $2^{-128}$ to be successful. In addition, the message cannot be distinguished from a randomly generated value due to the strength of the encryption algorithm with respect to chosen plaintext attacks. As the message cannot be decrypted, the identity remains hidden and, due to the usage of an authenticated encryption mode, the integrity is also provided. The authentication is offered in the next timeslot as the hash in the encrypted message can then be validated. Only the node with the correct prestored information is able to derive this hash. □

**Theorem** **2.**
*The protocol is able to offer integrity, authentication, and nonrepudiation with a minimum security level of 128 bits in case there is at most one malicious node in the network.*


**Proof.** If there is one malicious node in the network, it is able to derive the encryption key used to encrypt the messages of the nodes and the gateway and, thus, break the confidentiality and the anonymity of the messages. However, this node possesses no further information to break the authenticity of the messages from either the gateway or the other nodes, since it requires knowledge of prestored information to derive $K_{i}^{j1}$, which is only available in the other nodes and gateway due to the unique construction of *b* on which $K_{i}^{j1}$ is computed. Due to the strength of the hash function with respect to preimage attacks, it is impossible to find a collision on the hash function. □

**Theorem** **3.**
*The protocol is not able to offer integrity, authentication, and nonrepudiation with a minimum security level of 128 bits in at most $\frac{2^{m}}{2^{n}-1}\%$ of nodes per timeslot in the case where there are m malicious nodes collaborating in the network.*


**Proof.** If there are *m* malicious nodes collaborating in the network, they are able to construct the security material of all nodes with identity *i* for which b=(i+j mod $(2^{n}-1))+1$ can be constructed as a linear combination of the values $m_{i}+j$ mod $(2^{n}-1))+1$, with $m_{i}$ being the identities of the malicious nodes. In total, for each timeslot, there are at most $2^{m}-m-1$ other nodes besides the malicious nodes involved. Once the security material is revealed, the security features of these nodes are not addressed anymore.However, note that due to the construction of the key material, which is divided in a circular way, the infected $2^{m}-m-1$ nodes will vary over the different and independent timeslots. Due to the nonrepudiation feature, the source of the attack can be easily retrieved. □

With respect to the resistance against Denial of Service (DoS) attacks, we notice that it is resistant as long as the nodes are not malicious. Following the attacker model (second point), the gateway is considered to be trusted, so it is impossible that $K_{0}^{j+1}$ is leaked by assumption. It is also impossible to derive another value of $K_{0}^{j+1}$ for an outsider, which might be valid (because of Theorem 1). Moreover, there are two checks that need to be done, the one on $K_{0}^{j+1}$ and $K_{1}^{(j+1),1}$. Even in the extremely unlikely case the attacker knows the value $K_{0}^{j+1}$, it should also know the exact value of ${\bar{K}}_{e}^{j}$ since the nodes can already do the decryption; if it is useless, it may assume that the message is invalid. If ${\bar{K}}_{e}^{j}$ is leaked, which is due to the strength of the encryption algorithm only possible by means of exhaustive search, a DoS attack becomes possible. Therefore, malicious insider nodes can still overload the network, but this is a problem that is very difficult to overcome.

## 6. Performance Evaluation

### 6.1. Computational Cost

We focus on the side of the nodes in the network and will determine the computational efforts for each of the four steps to be performed as the reaction of the nodes on reception of the broadcast message of the gateway (cf. Section 4.2.2). We concentrate here on the most costly cryptographic operations, these being encryption and hash. The cost for each of the fours steps is equal to the following:**Step****1:** 1 hash operation with input 256 bits and output length 128n+256 bits.**Step****2:** 1 decryption operation on 128 bit message, 1 decryption operation on a message with length $256+|M^{j}|$ bits, and 1 encryption on a message with bit length $n+256+|M_{i}^{j}|$.**Step****3:** $n_{r}$ decryptions of messages with bit length $n+256\;+\;|M_{i}^{j}|$, where $n_{r}$ is the number of nodes from whom the node wants to derive the message (neighbors, children in routing scheme) and verify the authenticity.**Step****4:** 1 hash operation on a message with length $256\;+\;|M_{i}^{j}|$, output 256 bits; $n_{r}$ hash operations on messages with input a message with length $256\;+\;|M_{i}^{j}|$ bits, output 256 bits.

As a proof of concept, we implemented our protocol on a constrained device, the Zolertia RE-Mote development board, which contains a CC2538 microcontroller using the ARM Cortex-M3 architecture. The device possesses 512 kB flash memory and has 32 kB RAM.

The reason why we only used this board for our measurements is because the behavior of the cryptographic primitives used is the same for the different ARM Cortex-M architectures, as shown for example in [19]. Timing measurements are made using the Data Watchpoint and Trace (DWT) unit of the microcontroller. This unit contains a monotonically increasing counter, which increases at the same frequency as the core of the microcontroller. The error on the measurements is, thus, negligible.

Figure 2 shows the time it takes for an individual node to perform the required security actions in case it has to check $n_{r}$ neighboring nodes. As a consequence, we can conclude that it takes less than 0.18 s if every node has less than 5 neighboring nodes to verify, while it takes less than 6 s if every node has to check the input of 250 neighboring nodes. As a consequence, this performance shows that the protocol is feasible for realistic settings.

### 6.2. Communication Cost

Denote the average length of the message $M^{j},M_{i}^{j}$ by |M|. During each timeslot, the node receives a message of length $\left(n_{r}+1\right)\log_{2}t+\left(2+n_{r}\right)\cdot 256+n_{r}\cdot n+\left(n_{r}+1\right)\left|M\right|$. The node sends, during each timeslot, its own message of length $\log_{2}t+n+\left|M\right|+256$ and forwards the message of its $n_{r}$ children of the same length. There are no additional intermediate messages needed in the protocol to define separate session keys. Figure 3 shows the total number of sent and received bytes in case the length of the transmitted message of all nodes equals 32 bytes for a varying number of neighboring nodes. If a node has less than 5 neighboring nodes, 229 bytes will be sent and 445 bytes received over the channel. This demonstrates again that the overhead is reasonable for realistic scenarios.

### 6.3. Storage Cost

The storage cost is an inherent issue of the TESLA protocol. In order to still show the feasibility of our approach, we now provide a concrete example. The storage cost on each node mainly depends on the length of the hash chain and, thus, also the number of timeslots *t* in which authenticated messages can be sent. Since there is no relation between the individual values, there is no smart way to optimize the storage as can be typically done for a hash chain. Therefore, the storage size is dominated by 8t bytes. Considering a situation of 12 timeslots a day, corresponding with one every 2 h, the storage area requires 384 bytes per day or 140 kB per year, and thus, 420 kB for 3 years, which is feasible on a Zolertia RE-Mote given the fact that it contains 512 kB of flash memory.

## 7. Conclusions and Future Work

This paper proposes an effective and highly efficient scheme to enable broadcast authentication in a distributed wireless sensor network containing constrained devices. In particular, it represents the first symmetric-key-based solution to address authentication and nonrepudiation at the node level without separate key agreement sessions among the communicating nodes. Moreover, besides authentication and nonrepudiation, the proposed protocol is also able to offer confidentiality and anonymity without the presence of malicious nodes. Even with at most one malicious node, authentication and nonrepudiation still remain valid. The impact is also limited when there are more malicious nodes collaborating.

The proposed scheme is constructed by means of a multivector hash chain shared among the different nodes in the network. The scheme perfectly fits for applications enabling low-cost sensing and actuating, where trust and fault isolation are essential.

Due to the underlying construction of a hash chain relying on timeslotted communication, the scheme also possesses some inherent issues. First, while confidentiality and group-based authentication are obtained in the same timeslot, the actual node-based authentication can only be performed in the next timeslot. However, we believe that this is an acceptable trade-off between security and efficiency. Second, as there is no inherent structure in the key chains to be stored at the different nodes, the storage cost is proportional to the number of timeslots to which security can be offered. However, we have shown that the security lifetime for a popular lightweight node is still realistic.

Finally, we need to make sure that the memory of the nodes leaving the network is erased in order to avoid protection against collaboration of inside malicious nodes; thus, applications relying on limited changes in the network topology are more suitable.

In the future, from a theoretical point of view, we will investigate how patterns among malicious nodes evolve through the different noninfected nodes over the different timeslots. From a practical point of view, we will set-up a real application—e.g., for CO_2_ monitoring in a building—and study the impact of different network parameters on the efficiency of the scheme.

## Figures and Tables

**Figure 1 sensors-22-01403-f001:**
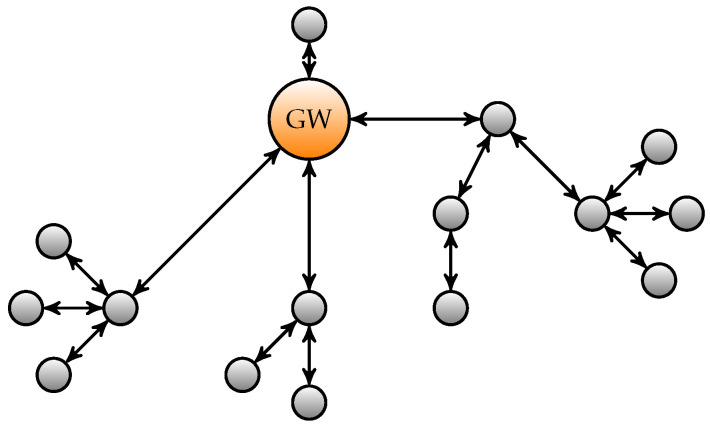
Representation of a WSAN. The orange node represents the gateway of the network. The other nodes represent sensors or actuators in the network.

**Figure 2 sensors-22-01403-f002:**
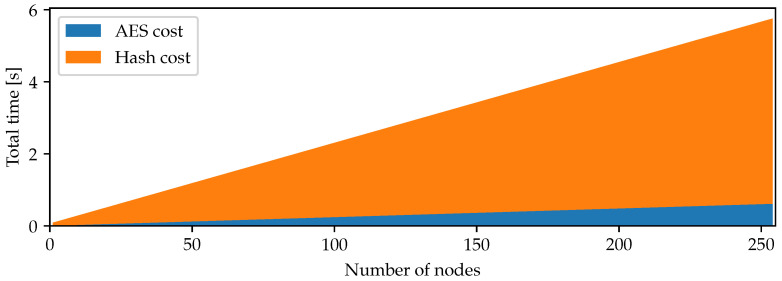
Time required for a Zolertia RE-Mote to perform the required security actions as a function of the number of neighboring nodes.

**Figure 3 sensors-22-01403-f003:**
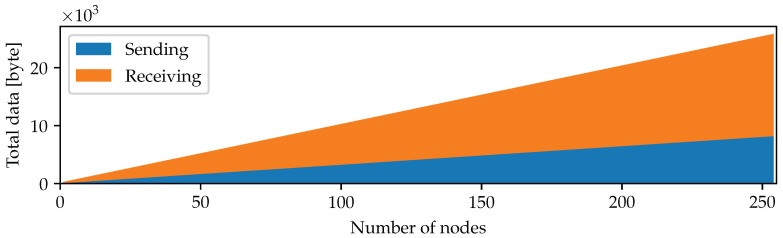
Amount of data sent and received during communication as a function of the number of neighboring nodes. Here, we assume the length of the transmitted messages to be equal to 32 bytes.

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
