# Peer review of "Symmetric-Key-Based Authentication among the Nodes in a Wireless Sensor and Actuator Network"

_sensors, 2022, doi:10.3390/s22041403_

Round 1
Reviewer 1 Report
The authors present a strategy for adding shared key -based authentication to WSAN networks without using a group key approach. Instead, they followed an approach close to TESLA, where communications are slotted and the receiver gets in each slot the key to validate the message received in the slot before.
The algorithm is very hard to read, fundamentally due to the writing style. More, the rationale for the construction of the keys stored in each node (beginning of page 5) is not given. What exactly is pretended with this special key computations.
The paper does not address a problem than can be created by assuming the attacker model given in 3.2: wrong messages can overflow the network, once an attacker knows the magic value of K_0^j+1. In fact, this value is what allows a node to keep a message broadcast by the GW until validating it on the next time slot. However, if two or more messages carry this exact same value in the same time slot, the receiver is not able to decide which one is valid or not until the next slot. As a consequence, an attacker can create a Denial of Service attack.
The proposed system as some natural limitations that are overlooked by the authors. The first one, is that messages are limited to one per slot. The other one, is that nodes that are not able to receive a message in slot j from the GW cannot communicate in that slot. And finally, and not less problematic, the key chains need to be refreshed from time to time. The longer the time, the more space is required on the nodes.
Another problem of the proposed schema is that it has no way to be fixed if a set of nodes gets compromised without reinstalling all the key chains. Therefore, the following sentence makes no sense:
"In case more than one node is compromised, the authentication and integrity of the gateway is still valid, but they are impacted for a limited number of nodes, which vary in the different timeslots. Therefore, it also enables fast fault isolation and effective intrusion detection and avoidance."
In fact, what the system provides is intrusion prevention, or avoidance, if an attacker could not derive keys used by other nodes in certain slots, but not detection. In fact, if node A is able to impersonate node B in slot j, this means that we cannot distinguish them, and therefore intrusion detection is impossible, unless a node keeps all messages received from the same origin in a given slot and more than one are valid for that slot. But, again, that takes us back to the DoS issue previously referred.
The re-keying of nodes is not addressed. This is a normal requirement of group keys, that should be refreshed once a new member enters or leaves the group. VANETs is an example of such an environment, because the vehicles can join and leave network groups quite dynamically, they are not required to be owned by the same entity but rather to be on a given location. But fast re-keying in such networks is challenging, as would be in your environment. For that purpose, you could benefit from insights taken from these articles, for example:
Veltri, Luca, et al. "A novel batch-based group key management protocol applied to the internet of things." Ad Hoc Networks 11.8 (2013): 2724-2737.
Guo, Ming-Huang, et al. "Centralized group key management mechanism for VANET." Security and Communication Networks 6.8 (2013): 1035-1043.
Cirne, Pedro, André Zúquete, and Susana Sargento. "TROPHY: Trustworthy VANET routing with group authentication keys." Ad Hoc Networks 71 (2018): 45-67.
Author Response
The authors present a strategy for adding shared key -based authentication to WSAN networks without using a group key approach. Instead, they followed an approach close to TESLA, where communications are slotted and the receiver gets in each slot the key to validate the message received in the slot before.
The algorithm is very hard to read, fundamentally due to the writing style. More, the rationale for the construction of the keys stored in each node (beginning of page 5) is not given. What exactly is pretended with this special key computations.
Response: We acknowledge that the reading is not straightforward. Therefore, we also added the examples to describe a concrete setting and to facilitate the understanding.
We appreciate very much your comment as it stimulated us to further investigate how to increase the readability. Following your suggestion to elaborate more on the reasoning of some of the decisions, we added the following text:
“The hash chain of key $K^0$ will be mainly used to derive material for authentication purposes and the hash chain of key $K_e^0$ will define the encryption keys in the process. There are $n$ independent components needed in the hash chain of $K^0$ in order to be able to define unique authentication material for each of the $2^n-1$ nodes in the network.”
“The key $K_i^{j1}$ is used in the later protocol to firstly verify the authentication of the hash chain of the gateway and to secondly authenticate its own message, which can be publicly verified by the other nodes in the next timestamp. The key $K_i^{j2}$ enables the derivation of the decryption key of the encrypted message coming from the gateway, which is also used as group encryption key by the nodes in the same timestamp. Note that this key $\overline K_e^j$ can only be defined after reception of the value $K_0^{j+1}$ as this is required to decrypt $K_i^{j2}$.”
Moreover, we also used bold notation in the definition of the key chain (line 156) in order to better visualize the path of the chain.
The paper does not address a problem than can be created by assuming the attacker model given in 3.2: wrong messages can overflow the network, once an attacker knows the magic value of K_0^j+1. In fact, this value is what allows a node to keep a message broadcast by the GW until validating it on the next time slot. However, if two or more messages carry this exact same value in the same time slot, the receiver is not able to decide which one is valid or not until the next slot. As a consequence, an attacker can create a Denial of Service attack.
Response:
Following the attacker model (second point), the gateway is considered to be trusted, so it is impossible that K_0^j+1 is leaked by assumption. It is also impossible to derive another value of K_0^j+1 for an outsider, which might be valid (because of Theorem 1). Moreover, there are two checks that need to be done, the one on K_0^j+2 and K_1^ (j+1),1.
Even in the extremely unlikely case the attacker knows the value K_0^j+1, it should also know the exact value of \overline K_e^2 since the nodes can already do the decryption, and if it is useless, it may assume that the message is invalid.
Therefore, there are no DoS attacks possible.
The proposed system as some natural limitations that are overlooked by the authors. The first one, is that messages are limited to one per slot.
Response: Multiple messages can in fact be securely sent in one timeslot using the same key material. We have clarified this at Step 2 of the protocol.
“Note that using the same key material, multiple messages can be sent by the nodes during this timeslot. An enumeration of the order can be included in the message.”
The other one, is that nodes that are not able to receive a message in slot j from the GW cannot communicate in that slot.
Response: The nodes can securely (confidentiality) communicate in the same slot (see step 2), but the final decision on the authentication can only be done in the next timeslot (see step 3).
There is no other scheme in literature that allows this authentication by means of symmetric key cryptography on node level (only at group level) and this is a unique feature of our proposed protocol. There is indeed this cost of one time slot delay in complete acceptance of the authentication. However, we think this is an acceptable trade-off between efficiency and security to be made.
In order to clarify it more, we have added the following text in the conclusion:
“Due to the underlying construction of a hash chain relying on timeslotted communication, the scheme also possesses some inherent issues. First, while confidentiality and group based authentication are obtained in the same timeslot, the actual node-based authentication can only be done in the next timeslot. However, we believe that this is an acceptable trade-off between security and efficiency. Second, as there is no inherent structure in the key chains to be stored at the different nodes, the storage cost is proportional with the number of timeslots to which security can be offered. However, we have shown that the security lifetime for a popular lightweight node, is still realistic.”
And finally, and not less problematic, the key chains need to be refreshed from time to time. The longer the time, the more space is required on the nodes.
Response: Key refreshment is indeed the major disadvantage. It is again inherent at the TESLA scheme. We explain in Section 6.3 that the storage cost is reasonable for a concrete scenario. We have added an additional sentence to explicitly acknowledge this issue and also added the text mentioned above in the previous response.
Another problem of the proposed schema is that it has no way to be fixed if a set of nodes gets compromised without reinstalling all the key chains. Therefore, the following sentence makes no sense:
"In case more than one node is compromised, the authentication and integrity of the gateway is still valid, but they are impacted for a limited number of nodes, which vary in the different timeslots. Therefore, it also enables fast fault isolation and effective intrusion detection and avoidance."
Response: Authentication of the gateway will always be possible as we have assumed in the attack model that the gateway is trusted and we also assume that the hash functions are resistant for collision and pre-image attacks. The only problem is that when more than one node is compromised and when they collaborate, then the authentication of some of the other nodes is not trusted anymore. This follows from the fact that the authentication keys of these nodes can be revealed, in particular the nodes who possess the key material that correspond with a linear combination of the “b’s” of the captured nodes. However, these affected nodes vary for the different timestamps (see Theorem 3).
In fact, what the system provides is intrusion prevention, or avoidance, if an attacker could not derive keys used by other nodes in certain slots, but not detection. In fact, if node A is able to impersonate node B in slot j, this means that we cannot distinguish them, and therefore intrusion detection is impossible, unless a node keeps all messages received from the same origin in a given slot and more than one are valid for that slot. But, again, that takes us back to the DoS issue previously referred.
Response: As the construction of the key K_i^j1 relies on the parameter b, which is determined by both the node i and the timeslot j, a coalition of malicious nodes will never be able to impersonate one specific node over multiple consecutive timeslots.
Instead, if the gateway is able to retrieve the identity of the malicious nodes and to communicate this to the rest of the nodes, the nodes in the network can then derive the different nodes for which the input may be corrupted by the malicious nodes for each specific timeslot. See also Theorem 3.
As mentioned in future work, the evolution of the patterns among the malicious nodes will be the subject of further investigation.
The re-keying of nodes is not addressed. This is a normal requirement of group keys, that should be refreshed once a new member enters or leaves the group. VANETs is an example of such an environment, because the vehicles can join and leave network groups quite dynamically, they are not required to be owned by the same entity but rather to be on a given location. But fast re-keying in such networks is challenging, as would be in your environment. For that purpose, you could benefit from insights taken from these articles, for example:
Veltri, Luca, et al. "A novel batch-based group key management protocol applied to the internet of things." Ad Hoc Networks 11.8 (2013): 2724-2737.
Guo, Ming-Huang, et al. "Centralized group key management mechanism for VANET." Security and Communication Networks 6.8 (2013): 1035-1043.
Cirne, Pedro, André Zúquete, and Susana Sargento. "TROPHY: Trustworthy VANET routing with group authentication keys." Ad Hoc Networks 71 (2018): 45-67.
Response: Our protocol fits in particular distributed WSANs with broadcast communication, consisting of constrained sensors (see Network model 3.1). Changes in the network, nodes addition and removal are limited. In particular, in case of node removal, the memory on the node should be erased.
This is a completely different setting as VANETs, which deal with dynamic construction of group keys. These schemes do not consider authentication of node level in the group, but only group authentication. Moreover, in the case of VANETs, the nodes are not constrained and so more compute intensive mechanisms based on public key cryptography can be applied. The three proposed works on VANETS all include public key based operations to ensure the authentication.
Note that the addition and removal of other nodes in the network is addressed in Section 4.2.3.
In order to stress this difference in network model, the following sentence has been added to the network model.
“We assume that changes in the network are limited. Nodes can be added, but if nodes are removed, then also the memory on it should be erased.”
We also added in the conclusion:
“Finally, we need to make sure that the memory of the nodes leaving the network is erased in order to avoid protection against collaboration of inside malicious nodes, and thus applications relying on limited changes in the network topology are more suitable.”

Reviewer 2 Report
- On line 274, the multiplication notation is incorrect.
- The authors used a CC2538 microcontroller and Zolertia RE-Mote development board. On what basis did the authors choose these models? I think it would be good to describe what was the reason for choosing this microcontroller and this development board.
- It is advisable to use a different microcontroller to compare the performance of both devices.
- How large are the calculation errors (numerical errors)? What influences these errors? Are there significant variations in the results for this reason, and how is the proposed network dealing with this?
- Have the authors considered problems with wireless transmission? What types of problems could be? Does the proposed network have security features against these problems?
- Conclusions, I believe, is way too short. It is necessary to summarize the advantages and disadvantages of the proposed solution, preferably on the basis of the solutions described in the introduction.
- Please correct the article formatting to conform to the Sensors requirements.
Author Response
- On line 274, the multiplication notation is incorrect.
Response: Thank you for this comment. This is now fixed.
- The authors used a CC2538 microcontroller and Zolertia RE-Mote development board. On what basis did the authors choose these models? I think it would be good to describe what was the reason for choosing this microcontroller and this development board.
Response: We only used the Zolertia RE-Mote development board, which uses the CC2538 as its microcontroller. This development board is the one we use at our research lab since the focus of the board is wireless communication. The ARM Cortex-M architecture is the most used microcontroller architecture in IoT. From the paper about the performance of SHA3 on constrained devices we see that the behaviour is quasi the same for all Cortex-M achitectures.
We added this motivation in the paper.
« The reason why we only used this board for our implementation tests is because the 297
behaviour of the cryptographic primitives used is the same for the different ARM Cortex-M 298
architectures, as shown for example in [21]. »
- T. Vandervelden, R. De Smet, K. Steenhaut, A. Braeken, SHA3 and Keccak variants computation speeds on constrained devices, 384, Future Generation Computer Systems, 128, pp. 28-35, 2022.
- It is advisable to use a different microcontroller to compare the performance of both devices.
Response: See motivation in previous point.
- How large are the calculation errors (numerical errors)? What influences these errors? Are there significant variations in the results for this reason, and how is the proposed network dealing with this?
Response: We assume the connection between the devices does not fail, thus assuming that every packet arrives. This approach is typically done in the discussion of new security protocols. The measurements were made using the Data Watchpoint and Trace (DWT) unit. This unit has a counter that increments monotonically and has the same period as the core of the microcontroller. Thus the error on the measurements is negligible. We have now specified this in the paper.
« Timing measurements are made using the Data Watchpoint and Trace (DWT) unit of the microcontroller. This unit contains a monotonically increasing counter, which increases at the same frequency as the core of the microcontroller. The error on the measurements is thus negligible.»
- Have the authors considered problems with wireless transmission? What types of problems could be? Does the proposed network have security features against these problems?
Response: As in the previous point we assume that the connection between devices does not fail. We consider this type of evaluation out of scope for our paper. The main goal of the paper is to present a new security protocol and to demonstrate its feasibility by means of a proof of concept implementation, which is independent of the underlying network problems. It is indeed a good suggestion to include as future work.
« In the future, from a theoretical point of view, we will investigate how patterns among malicious nodes evolve through the different other non-infected nodes over the different timeslots. From a practical point of view, we will set-up a real application, e.g. for CO2 monitoring in a building, and study the impact of different network parameters on the efficiency of the scheme. »
- Conclusions, I believe, is way too short. It is necessary to summarize the advantages and disadvantages of the proposed solution, preferably on the basis of the solutions described in the introduction.
Response: Indeed, based on inputs of the other reviewers, we have added important insights into the conclusion.
- Please correct the article formatting to conform to the Sensors requirements.
Response: We are sorry for this. We believed that we applied the correct formatting. Please let us know which adaptations are required.

Round 2
Reviewer 1 Report
The authors addressed most of the remarks, but did not understand the effective risks of DoS.
As the authors say, in each slot many messages may be received. How many? Well, it deppends on the receivers' buffers, because messages can be tentatively accepted based on the validation of part of the message, and properly accepted on the next slot.
However, an eavesdropper can learn the magic tag that makes a message tentatively accepted on slot j (K_0^j+1), before being properly validated on slot j+1 (the message is decrypted on slot j but validated on slot j+1). Thus, the eavesdropper, once knowing that magic tag, can flood the receivers' buffers with garbage, possibly preventing legitimate messages to be received. Thus, the system is an easy target for DoS and that must be properly acknowledged in the paper.
Note that if you add some redundancy to M^j so that a note could drop fake messages received by a DoS attacker, a compromized node could still do it, because it would know the key K_e^j that it used to decrypt the message.
Author Response
The authors addressed most of the remarks, but did not understand the effective risks of DoS.
As the authors say, in each slot many messages may be received. How many? Well, it deppends on the receivers' buffers, because messages can be tentatively accepted based on the validation of part of the message, and properly accepted on the next slot.
Response: That is correct. Thank you for this suggestion. We have added this to the paper.
“The amount of messages that can be received in each slot depends on the receivers’ buffer, because messages are tentatively accepted based on the validation of the first part of the message, and completely accepted on the next slot if the authentication tag is correct.”
However, an eavesdropper can learn the magic tag that makes a message tentatively accepted on slot j (K_0^j+1), before being properly validated on slot j+1 (the message is decrypted on slot j but validated on slot j+1). Thus, the eavesdropper, once knowing that magic tag, can flood the receivers' buffers with garbage, possibly preventing legitimate messages to be received. Thus, the system is an easy target for DoS and that must be properly acknowledged in the paper.
Note that if you add some redundancy to M^j so that a note could drop fake messages received by a DoS attacker, a compromized node could still do it, because it would know the key K_e^j that it used to decrypt the message.
Response: In order to learn the magic tag, an exhaustive search on the encryption key \overline K_e^j is required. It is true that a malicious insider node has the possibility to do a DoS attack, but this is impossible to overcome and you have in other schemes too. In any case, it is a good remark to be added to the paper. Therefore, we have added to the security analysis:
“With respect to the resistance against Denial of Service (DoS) attacks, we notice that it is resistant as long as the nodes are not malicious.
Following the attacker model (second point), the gateway is considered to be trusted, so it is impossible that $K_0^{j+1}$ is leaked by assumption. It is also impossible to derive another value of $K_0^{j+1}$ for an outsider, which might be valid (because of Theorem 1). Moreover, there are two checks that need to be done, the one on $K_0^{j+1}$ and $K_1^{(j+1),1}$.
Even in the extremely unlikely case the attacker knows the value $K_0^{j+1}$, it should also know the exact value of $\overline K_e^j$ since the nodes can already do the decryption, and if it is useless, it may assume that the message is invalid. If $\overline K_e^j$ is leaked, which is due to the strength of the encryption algorithm only possible by means of exhaustive search, a DoS attack becomes possible. Therefore, malicious insider nodes can still overload the network, but this is a problem which is very difficult to overcome.”
Reviewer 2 Report
Please see the page for the author guidelines.
https://www.mdpi.com/journal/sensors/instructions
Author Response
Please see the page for the author guidelines.
Response: We carefully checked the author guidelines. In particularly, we completed the acknowledgment section now. We hope it is fine now.